# Exploring Interventions for Sleep Disorders in Adolescent Cannabis Users

**DOI:** 10.3390/medsci6010011

**Published:** 2018-02-08

**Authors:** Tzvi Furer, Komal Nayak, Jess P. Shatkin

**Affiliations:** Department of Child and Adolescent Psychiatry, Child Study Center at Hassenfeld Children’s Hospital of New York at NYU Langone, One Park Avenue, 7th Floor, New York, NY 10016, USA; Komal.Nayak@nyumc.org (K.N.); Jess.Shatkin@nyumc.org (J.P.S.)

**Keywords:** adolescent, cannabis, sleep, interventions

## Abstract

This review summarizes the available literature on the intersection of adolescent cannabis use and sleep disturbances, along with interventions for adolescent cannabis users who suffer sleep impairments. Adolescents are susceptible to various sleep disorders, which are often exacerbated by the use of substances such as cannabis. The relationship between cannabis and sleep is bidirectional. Interventions to improve sleep impairments among adolescent cannabis users to date have demonstrated limited efficacy, although few studies indicating the benefits of behavioral interventions—such as Cognitive Behavior Therapy for Insomnia or Mindfulness Based Stress Reduction—appear promising in the treatment of sleep disorders, which are present for users of cannabis. Further research is necessary to elucidate the precise mechanisms by which cannabis use coexists with sleep impairments, along with effective interventions for those users who suffer sleep difficulties.

## 1. Introduction

Adolescents undergo tumultuous changes in sleep. By early puberty, the majority of adolescents begin to show preference for later sleep and wake times [1,2], with evident symptoms of sleepiness during the daytime [3]. Several studies have demonstrated that a substantial proportion of adolescents are sleep deprived [3], which often leads to irregular sleep schedules between weekdays and weekends [1,4]. For example, 17 percent of adolescents report trouble falling asleep before 2 a.m. at least three times per week [5]. This pattern of disrupted sleep often results in circadian misalignment, with roughly one to seven percent of adolescents meeting diagnostic criteria for delayed sleep phase disorder, or DSPD [6,7,8]. DSPD itself typically leads to a series of negative outcomes, such as academic difficulties, irritability, and depressed mood. Delaying the middle school and, especially, high school start times is a proven intervention to not only reduce sleepiness, fatigue, and negative mood, but also improve academic performance and even reduce automobile accidents among those teens who drive [2]. The U.S. Centers for Disease Control (CDC) has, in fact, recommended that high schools not start before 8:30 a.m.; however, only 20 percent of schools in the U.S. adhere to this recommendation [2]. 

Adolescence is also a time of increased risk-taking behavior, including experimentation with alcohol, cannabis, and tobacco [9]. According to studies, cannabis is the most prevalent psychoactive substance used by adolescents in the United States [10]. Not surprisingly, irregular sleep schedules, daytime sleepiness, and frequency of substance use have been found to correlate with one another [11]. Multiple studies [12,13,14] demonstrate that sleep problems precede, and are often predictive of, future adolescent substance use. Consequently, the extant literature on substance use and sleep indicates that the effect is bidirectional, with both a lack of sleep and continued use of substances each leading to potentially more difficulties with the other [15].

Cannabis is the most common substance found in drug screens following adolescent arrests, emergency room admissions, and autopsies [16]. As of September 2017, 28 states and the District of Columbia have legalized cannabis for medical purposes within the United States, and 7 states and the District of Columbia have legalized cannabis for recreational use; in addition, several other states have proposed bills to legalize cannabis within the next few years [17]. Surveys done in states where cannabis has been legalized indicate no significant trends in increased marijuana use post-legalization, with noted decreases in cannabis use within the past 30 days for both 10th grade students in Washington as well as overall youth use in Colorado [18]. In Alaska, the percentage of all high school students using cannabis use has remained stable, even post-legalization of marijuana [18]. As of 2015, 81% of 12th grade students across the United States cited being able to access marijuana easily if they desired, while 64% of 10th grade students and 35% of 8th grade students also reported “easy” access [9]. 

The annual prevalence of marijuana use has largely declined from 2013 to 2016, showing a drop from 12.7% in 2013 to 9.4% in 2016 for 8th grade students while use in 10th grade students dropped from 29.8% in 2013 to 23.9% in 2016 [9]. For 12th grade students, however, the annual prevalence has held mostly steady, with annual prevalence noted as 36.4% in 2013 while documented as 35.6% in 2016. General attitudes of students have trended towards greater acceptance of marijuana use, with the perceived risk of regular cannabis use noted in 2016 as 57.5, 44, and 31% for 8th, 10th, and 12th graders, respectively. Meanwhile, perceived risk of regular cannabis use was previously greater in 2013, noted as 61, 46.5, and 39.5% for 8th, 10th, and 12th graders, respectively [9]. 

In this paper, we review the current literature on sleep disorders in adolescents who use cannabis, the bidirectional effects of cannabis and sleep disorders, and behavioral interventions for adolescents who use cannabis and have comorbid sleep disorders. 

## 2. Cannabis and Sleep

It is reported that the cannabis plant has over 100 cannabinoids, of which are the active components of marijuana that work primarily on the endocannabinoid system. Delta-9-tetrahydrocannabinol (THC) and cannabidiol (CBD) are two of the most researched cannabinoids, and are classified as phytocannabinoids [17]. The stability of the circadian rhythm is impaired after administration of THC [19]. In response to chronic administration of THC, the body adapts to a state of reduced slow wave sleep [20]. THC is psychoactive via activation of specific CB1 (cannabinoid) receptors that regulate mood, appetite, and memory [21]. Activation of CB1 receptors creates the ‘high’ feeling reported by cannabis users, and is noted to have a biphasic effect corresponding to either high or low doses of THC. Animal model studies have defined this ‘biphasic’ difference based on the quantity of THC: specifically that low doses of THC produce an anxiolytic effect, while higher doses of THC can induce panic symptoms [22,23,24]. Use of THC overall can potentially produce either cognitive impairment or acute psychosis. CBD differs from THC in that it does not activate the CB1 cannabinoid receptors, and is therefore not ‘psychoactive’ [25]. CBD does not display the same intoxicating effects as THC, and is currently being researched as an adjunctive treatment for symptoms of psychosis [25]. On a molecular level, the endocannabinoid system is presumed to be involved in the regulation of the circadian sleep–wake cycle [26] and the maintenance and promotion of sleep [20]. Overall, chronic cannabis use has been found to be associated with poorer sleep quality and disruptions in sleep [27]. It is generally accepted, however, that a bidirectional effect exists between regular cannabis use and sleep impairments, although the scientific literature is limited. 

It has been shown that acute exposure to cannabis can reduce sleep latency, overall time in rapid eye movement (REM) phase, and decreased REM density [28], while administration of CBD and/or THC also has profound effects on sleep. The relationship of cannabidiol with sleep has had different noted outcomes; low-dose CBD has been found to create a stimulating effect, while high dose CBD has shown to have a sedating effect [17]. Other studies of CBD have been mixed; one study demonstrated that cannabidiol blocked anxiety-induced REM sleep suppression without an effect on NREM sleep [29], while other research found that CBD injections caused an increase in the total percentage of sleep in rats [30]. Meanwhile, administration of THC may promote sleep onset and decrease total REM sleep time. Both CBD and THC have a profound effect on sleep architecture in consistent cannabis users. CBD has been found to decrease stage 3 (N3) sleep (commonly known as slow-wave sleep) when used in conjunction with THC, while THC and synthetic THC preparations have been associated with decreased sleep latency [31]. Stage 3 (N3) sleep has been shown to be important in the consolidation of hippocampus-dependent declarative memories [32], whereas previously it was theorized that REM sleep played a major role in memory consolidation [33]. This finding explains previous reports indicating the adverse effects of cannabis on working memory and the ability to maintain information [34]. Thus, the noted effects on sleep architecture play a significant role in cannabis’ known effects on memory, especially in adolescents. Disturbed sleep is reported in 67–73% of adults and 33–43% of adolescents during a quit attempt [35,36,37] and can be seen up to 45 days post-cessation [38]. Prolonged use of cannabis followed by rapid cessation is known to result in sleep disruption and a significant REM rebound, characterized by an increase in dreaming [35,38,39,40]. 

Most available data on the effects of cannabis on sleep describes the adult population; however, emerging evidence suggests that adolescents experience these same changes in sleep [41]. Numerous studies have found a correlation between sleep duration [42], self-reported sleep problems [43], and insomnia [13] among those adolescents who use cannabis. Sleep-timing characteristics are associated with substance use [44], with demonstrated differences between individual chronotypes. Chronotypes are specific behaviors indicative of an individual’s circadian rhythm, with morning chronotypes found to have earlier sleep onset and wake-up times, while evening chronotypes show later sleep onset patterns and more significant substance use behaviors [45]. 

Sleep also appears to have an important role in both cessation from cannabis and relapse [36]. Up to 65% of cannabis users, in fact, have identified poor sleep as the primary reason for relapse during a prior attempt at cessation. Paradoxically, cannabis has been described by users as having both a soporific and a sleep disrupting quality [46]. This observation may be due to the cyclical nature of substance use. Individuals may begin using cannabis to aid in sleep initiation, but once tolerance is established, they may require higher doses to yield the same effect. This pattern of use then leads to increased sleep disturbances during abstinence from cannabis, resulting in further relapses [37]. It is important to note that little data is available regarding regular users in natural settings, who are not seeking treatment for cannabis addiction or sleep disorders [47]. 

Cannabis has also been used in the medical treatment of various sleep-related disorders, including obstructive sleep apnea (OSA) and REM behavior disorder [17]. Among patients suffering from OSA, the endocannabinoid oleamide and exogenous cannabinoid THC have been found to reduce apneic events by alleviating serotonin-mediated OSA symptoms [48]. One case series has observed that cannabidiol reduced symptoms of REM behavior disorder among four adults with Parkinson’s disease [49]. Finally, limited research on excessive daytime sleepiness has noted that patients who tested positive for THC (indicating recent cannabis use) were more likely to meet criteria for narcolepsy [50]. 

## 3. Interventions 

Overall, there is limited data describing useful interventions for adolescents who use cannabis and suffer from sleep problems. Pharmacological interventions do not typically target the cannabis abuse itself, and most of the current treatment literature focuses on addressing symptoms of withdrawal, among which are the various sleep disturbances. Most of these studies have focused on adult patients, and little has been demonstrated in a naturalized clinical setting. Not surprisingly, both zolpidem and benzodiazepines in general have been found useful in treating sleep disturbances associated with cannabis withdrawal. Gabapentin, N-acetylcysteine (NAC), and naltrexone have each been shown to induce some reductions in the use of cannabis and in the prevention of relapse in small samples [51]. Of greater utility and relevance for most practitioners, however, are the benefits of behavioral interventions for the treatment of sleep problems relating to cannabis use. Over the past decade, numerous behavioral strategies have emerged for the treatment of insomnia and related sleep difficulties in both adults and adolescents. Often grouped under the rubric of “cognitive behavioral therapy for insomnia (CBT-I)”, these various tools are, in fact, unique and variable in efficacy. Studies of CBT-I have shown improvement in sleep quality in both adults and adolescents with sleep disturbances [52,53,54,55].

Important behavioral interventions highlighted in this review include stimulus control therapy (SCT), sleep restriction therapy (SRT), bright light therapy, sleep hygiene education, cognitive therapy, behavioral relaxation, and mindfulness based stress reduction. These tools are part of an array of options available to address sleep problems, and provide unique benefits and applications. The goal of SCT is to enhance the association between the bed/bedroom and sleep. This objective is accomplished largely by reducing the amount of time spent awake in bed, in addition to removing all potentially alerting stimuli from the room (e.g., cell phone, lighted clocks, etc.) [56,57,58]. SCT is effective both for acute and chronic insomnia [59]. Sleep restriction therapy is commonly employed alongside SCT. In this case, the goal of treatment is to strengthen the homeostatic sleep drive by restricting the amount of time spent in bed to only that during which the individual is actually sleeping. This process involves a thorough and regular logging of sleep hours, and therefore can be a bit more labor intensive, but ultimately improves sleep efficiency markedly.

Another effective treatment for insomnia is bright light therapy, also known as heliotherapy or phototherapy. Bright light exposure works by resetting circadian rhythms, specifically in individuals with sleep-phase delays [4], with greater efficacy when applied during the morning hours [60]. Sleep hygiene education delivers basic information about sleep, providing practical advice on managing circadian rhythm, body temperature, napping, exercise, diet, caffeine, alcohol, tobacco, and drugs in an effort to improve sleep quality [61,62]. Cognitive therapy for insomnia treats sleep disturbances by restructuring maladaptive thoughts, beliefs, and attitudes. This method addresses faulty beliefs, unrealistic sleep expectations, and diminished perceptions about control of sleep [63]. Finally, behavioral relaxation/arousal reduction utilizes skills akin to those taught in mindfulness-based stress reduction (MSBR). These procedures focus on mindfulness-based exercises to reduce the physical and psychological tension associated with sleep difficulty. Procedures include practices such as meditation, deep breathing, progressive muscle relaxation, and biofeedback. Numerous studies have identified sleep improvements from these practices [64,65,66]. One study looked at a combination of the above behavioral interventions, structured as a six-session multicomponent treatment model to improve sleep and decrease the risk of relapse in an adolescent population of substance users [67]. Significant improvement was found in sleep efficiency, sleep onset latency, and total sleep time. Additionally, there was a decrease in substance problems found 12 months after the conclusion of the study. 

## 4. Conclusions 

Cannabis use and sleep impairments go hand-in-hand and share a bidirectional relationship with each strongly influencing the other. Although good behavioral and pharmacological strategies independently exist for the treatment of sleep disorders, targeted interventions for sleep impairments among those who regularly use cannabis are needed. Future studies should examine behavioral and psychopharmacological treatment models for sleep disturbances among adolescent marijuana users, who remain a difficult population to study. Sleep patterns are also undoubtedly affected by growth and development during adolescence, and the biological effects of cannabis may also vary throughout development. Since cannabis use typically begins in adolescence, identifying interventions that are effective, practical, and accepted by youth is essential to decrease the risk of long-term sleep disorders and other co-morbidities. In the meantime, practitioners are advised to rely primarily upon behavioral methods to improve the sleep of adolescents who use cannabis.

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
