# Peer review of "Exploring Interventions for Sleep Disorders in Adolescent Cannabis Users"

_medsci, 2018, doi:10.3390/medsci6010011_

Round 1

Reviewer 1 Report

The article is of great importance and timing.

I recommend just a few minor revisions:

78 - please use full term of "delta-9-tetrahydrocanabinol" before using the abbreviation of "THC" as on 80, please use CBD throughout the text after you introduced this abbreviation as on 78, please use full term "Rapid eye movement" before you use abbreviation REM as on 92.

88 - sentence "THC differs from cannabidiol,  " might be confusing to a reader, because you already mentioned that it does in 82. May be you meant "Cannabidiol differs from THC.."

I would like to emphasize the importance of the fact that  use of Cannabis in adolescence is neither recommended nor accepted,  and is illegal. Importance of treatment and search for  successful treatment interventions for Cannabis abuse in adolescents is essential. Use of behavioral treatment modalities for Cannabis-induced sleep disturbances described by the authors could be successfully integrated in clinical care of adolescents  with total sessation of Cannabis use as the ultimate goal of treatment. I would like to recommend that the authors mention the importance of sleep regulation as the integral part of care for Cannabis abusing  adolescents.

Author Response

Hi,

Thank you for taking the time out to review our submission and provide comments. We have been able to incorporate many of your suggested revisions, of which were helpful. 

-The paper was updated to use the full term for THC prior to its abbreviation later in the paper. 

-The sentence has been updated to create less confusion between THC and CBD. 

We appreciate the emphasis on cannabis and successful interventions.  We hope this is a platform for future research into the topic, especially regarding sleep regulation. 

Reviewer 2 Report

Overall this is a timely review.

The following statement (lines 12-16) in the Abstract is incomplete: " Interventions to improve sleep impairments among adolescent cannabis users to date have demonstrated limited efficacy, although few studies indicating the benefits of behavioral interventions, such as Cognitive Behavior Therapy for Insomnia or Mindfulness Based Stress Reduction." Did you wish to add something at the end like "...although a few studies...such as...CBTi....appear encouraging" ?

The order of references is confusing since it neither follows a numbering pattern by order of first appearance in the text, nor is it alphabetically arranged.   

References are needed to support statements made in lines 33, 35, 52.

The entire paragraph from line 56-64 references the publication by Jonhnston LD et al. The title of this report from Univ of Michigan itself suggests the data analyzed was from 1975-2015. Therefore, the authors cannot be accurate in projecting this data to 2016 as they do in this paragraph! Further, in line 59: the word "then" is not correct for the sentence structure.

Line 94: sentence needs to be re-written. The words " has been studied to" should be replaced to make a better sentence.

References 44 and 45  in lines 116-119 pertain to alcohol use. Please provide citations for cannabis related works or avoid generalizing form alcohol to cannabis unless you are sure this can be extrapolated.

Section 2 on Cannabis and Sleep (lines 71- 136) has a lot of material but is a little difficult to read in its current format for an unfamiliar reader. The flow is not smooth and seems to jump around. The presented material will be more effective if rearranged in a logical and stepwise fashion. For example,  THC and cannabidiol related research may be first introduced by highlighting the two compounds and their receptors. Then one may lead the reader in a step wise fashion through the effects of cannabis on sleep at the molecular level, through various aspects of sleep and circadian rhythms from micro to macro architecture. Consider "the effect of cannabis on sleep" and "effect of sleep on cannabis" can be two subheads to point the reader in the intended direction and improve readability. If editorially acceptable, a table showing key take-away points will be very useful for this complex data.  

Author Response

Hi,

Thank you for taking the time out to review and comment on our paper.  Your suggestions were valuable, and we've made a number of revisions that reflected your comments. 

-The initial abstract was updated and now reads completely. 

-We have extensively updated the references to go into sequential order as we deleted a portion of our paper previously, and have now fixed it accordingly. 

- References have also been updated in the suggested areas, and have cited references. 

-I have updated several references that were incorrectly labeled (including the Monitoring the Future Study which was published in 2017 not 2016), so the references are no more accurate.  Additionally, the references pertaining specifically to alcohol use were removed and replaced with more accurate 

-Where appropriate, we have updated the grammar and sentence structure so it that it flows better. 

-We have restructured Section 2 entirely so that it flows better and reads in a more step-wise fashion.   We did not want to change much about the structure, but we believe that it proceeds in a more appropriate manner. 

Again, thank you for your highly appreciated comments which allowed us to improve the paper significantly.